



# Why we need radar, lidar, and solar radiance observations to constrain ice cloud microphysics

Florian Ewald[1], Silke Groß[1], Martin Wirth[1], Julien Delanoë[2], Stuart Fox[3], and Bernhard Mayer[4]

[1]Deutsches Zentrum für Luft und Raumfahrt, Institut für Physik der Atmosphäre, Oberpfaffenhofen, Germany
[2]LATMOS/UVSQ/IPSL/CNRS, Guyancourt, France
[3]Met Office, FitzRoy Road, Exeter, EX1 3PB, UK
[4]Meteorologisches Institut, Ludwig Maximilians Universität, München, Germany

**Correspondence:** Florian Ewald (florian.ewald@dlr.de)

**Abstract.** Ice clouds and their effect on Earth's radiation budget are one of the largest sources of uncertainty in climate change predictions. The uncertainty in predicting ice cloud feedbacks in a warming climate arises due to uncertainties in measuring and explaining their current optical and microphysical properties as well as from insufficient knowledge about their spatial and temporal distribution. This knowledge can be significantly improved by active remote sensing, which can help to explore
the vertical profile of ice cloud microphysics, such as ice particle size and ice water content. This study focuses on the well-established variational approach VarCloud to retrieve ice cloud microphysics from radar-lidar measurements.

While active backscatter retrieval techniques surpass the information content of most passive, vertically integrated retrieval techniques, their accuracy is limited by essential assumptions about the ice crystal shape. Since most radar-lidar retrieval algorithms rely heavily on universal mass-size relationships to parameterize the prevalent ice particle shape, biases in ice water
content and ice water path can be expected in individual cloud regimes. In turn, these biases can lead to an erroneous estimation of the radiative effect of ice clouds. In many cases, these biases could be spotted and corrected by the simultaneous exploitation of measured solar radiances.

The agreement with measured solar radiances is a logical prerequisite for an accurate estimation of the radiative effect of ice clouds. To this end, this study exploits simultaneous radar, lidar, and passive measurements made on board the German High
Altitude and Long Range Research Aircraft. By using the ice clouds derived with VarCloud as an input to radiative transfer calculations, simulated solar radiances are compared to measured solar radiances made above the actual clouds. This radiative closure study is done using different ice crystal models to improve the knowledge of the prevalent ice crystal shape. While in one case aggregates were capable of reconciling radar, lidar, and solar radiance measurements, this study also analyses a more problematic case for which no radiative closure could be achieved. In this case, simultaneously acquired in-situ measurements
could narrow this inability to an unexpected high ice crystal number concentration.

## 1   Introduction

Ice clouds play an essential role in the climate system since they have a large effect on Earth's radiation budget, on heating and cooling rates throughout the atmosphere and on the water cycle (Liou, 1986). Thin ice clouds, so-called cirrus clouds, play a





special role in Earth's climate due to their semi-transparency for solar radiation. While cirrus reflect only a small portion of the

incoming solar radiation, they are very effective at inhibiting the transmission of thermal radiation from the surface and lower

troposphere into space due to their location in the upper troposphere where low temperatures prevail. Averaged globally, cirrus

clouds have thus a net warming effect on the earth–atmosphere system (Hong et al., 2016). The level of scientific understanding

if this effect of ice clouds will change in a warming climate including various cloud-climate feedbacks is, however, still low

(IPCC, 2013). Measurement uncertainties of their current optical and microphysical properties, as well as the insufficient

knowledge about their spatial and temporal distribution, are contributing to this problem (Eliasson, 2011). The solar radiative

effect of ice clouds is governed by their optical thickness and their particle size and shape (Eichler et al., 2009). It is therefore

essential to improve and validate current techniques to retrieve these cloud properties.

## 1.1 Active vs. passive remote sensing of ice clouds

Since the early days of cloud remote sensing from space, properties like cloud cover, optical thickness, effective radius, or total

water path were derived using bi-spectral retrieval techniques in the solar (Nakajima and King, 1990; Han et al., 1994; Platnick

et al., 2003) as well as thermal spectral range (Rossow et al., 1989; Ewald et al., 2013). Sub-pixel cloud inhomogeneity (Zinner

and Mayer, 2006), three-dimensional radiative effects (Marshak et al., 2006) and problematic viewing geometries (Cho et al.,

2015) can however cause significant biases when using these passive techniques. While passive microwave observations are

largely unaffected by these effects, uncertainties of the surface emissivity limit this technique from space to thicker ice clouds

(Zhao et al., 2002). Almost all of these challenges are tied to an uncontrolled light source, where either the origin or path

of the measured light is partly unknown. Active remote sensing techniques rely on their own light source and can therefore

significantly improve the remote sensing of cloud microphysics from space or aircraft. Time of flight measurements with pulsed

techniques such as radar or lidar can even yield profiles of cloud properties.

## 1.2 Combination of radar, lidar, and passive measurements

The combination of radar and lidar measurements can even provide height resolved information of ice cloud microphysics.

Since radar reflectivity $Z$ is proportional to the sixth moment of the particle size distribution (PSD), its measurement is highly

sensitive to the cloud particle size. In contrast, the lidar backscatter coefficient $\beta$ is linked to extinction $\alpha$ which is proportional

to the second moment of the PSD and, in turn, more sensitive to the cloud particle number concentration. Due to this different

sensitivity to particle sizes, both instruments complement each other in multiple ways. In the overlap region of both instruments,

two moments of the PSD (e.g. particle number concentration and particle size) can be determined. Furthermore, the lidar

contributes complementary measurements for optically thin ice clouds with a too weak radar backscatter, while the radar can

penetrate deep convective ice clouds with precipitation for which the lidar signal is quickly extinguished.

First steps towards combined radar-lidar retrievals were made by Intrieri et al. (1993), Donovan and van Lammeren (2001),

Tinel et al. (2005), and Mitrescu et al. (2005). While the extinction-to-backscatter ratio (lidar ratio $S$) had to be assumed in

the first approach, the latter studies already combined radar reflectivity $Z$ and attenuated lidar backscatter coefficient $\beta_a$ while

varying $S$. These methods were, however, only applicable to cloud regions with overlapping cloud and lidar measurements.





More recent approaches (e.g. Delanoë and Hogan, 2008) solved this limitation by using optimal estimation frameworks that fit a microphysical model profile to lidar and radar measurements.

For the Cloud-Aerosol Lidar with Orthogonal Polarization (CALIOP) aboard CALIPSO (Winker et al., 2010) and the Cloud
Profiling Radar (CPR) aboard CloudSat (Stephens et al., 2002), as well as for the upcoming ESA/JAXA EarthCARE mission (Illingworth et al., 2015), variational optimal estimation algorithms have been developed, which combine radar, lidar (e.g. 2C-ICE; Deng et al., 2010) and thermal radiance measurements (VarCloud; Delanoë and Hogan, 2008) in a unified framework. While the VarCloud algorithm is a versatile framework which is constantly developed further (Delanoë et al., 2014; Cazenave et al., 2019), a version called DARDAR (Delanoe and Hogan, 2010) is used to retrieve operational ice cloud microphysics from
CloudSat and CALIPSO.

Up to now, all of these methods rely heavily on radar-lidar profile measurements and only make limited use of vertically integrating measurements like thermal radiances. The incorporation of passive measurements in the solar spectrum is planned for the future unified algorithm CAPTIVATE, as proposed by Illingworth et al. (2015) for the EarthCARE mission.

### 1.3 Problem Statement

While combined radar-lidar retrievals excel most passive retrievals by providing full profiles of cloud properties, their performance is also hampered by limitations specific to the backscatter measurement technique. The height resolved retrieval of ice cloud microphysics is their unique feature, but the vertically integrated ice-cloud properties can be strongly biased since they are only constrained by thermal radiances (which saturate quite early) or not constrained by vertically integrating measurements at all. Most radar-lidar retrieval algorithms rely heavily on microphysical assumptions to retrieve two unknowns
from two measurements, e.g., ice water content and mean particle size. For this reason, they have to simplify the variability of naturally occurring ice crystals. The mass $M$ and projected area $A$ are commonly used properties to simplify the ice crystal variability since the radar reflectivity is proportional to $M^2$ and the lidar-extinction coefficient is proportional to $A$ (e.g. Delanoë et al., 2014). For that reason, large in-situ data sets are explored for relationships that associate ice particle sizes $D$ with their average, in-situ measured mass $M$ and projected area $A$ (e.g. Cazenave et al., 2019). Since these M–D and A–D relation-
ships change with particle shape, the performance of combined radar-lidar retrievals relies on the statistical representativeness of the sampled ice particle shapes in the used in-situ data.

Recent in-situ studies, however, found an extreme variability of m–D properties among clouds as well as within individual clouds volumes (Xu and Mace, 2016; Mace and Benson, 2017). They observed that the assumption of a constant M–D relationship (and thus constant shape assumption) can lead to a factor-of-2 uncertainty in ice water content retrievals. This finding
is consistent with numerous other studies that discovered large differences in IWC (up to a factor of 2) between different radar-lidar retrievals (Comstock et al., 2007; Zhao et al., 2012; Deng et al., 2012; Hong et al., 2016).

In many cases, these biases could already be identified during the remote sensing process when retrieved cloud properties disagreed with simultaneously acquired passive measurements. In this context, Stein et al. (2011) examined two different microphysical assumptions within the VarCloud retrieval framework: the standard ice crystal shape assumption of oblate spheroids
(following the M–D relationship of Brown and Francis, 1995) and a *bullet rosette* shape. In their study, Stein et al. (2011) could



show that optical depths are globally a factor-of-2 lower than those retrieved from MODIS when using oblate spheroids, but overestimated by the same factor when using the *bullet rosette* shape. This strong sensitivity to the ice crystal shape serves as motivation to use solar radiances as a valuable tool to obtain ice cloud microphysics with accurate optical properties. Moreover, solar radiation promises greater synergy with radar-lidar measurements compared to thermal radiation due to its deeper cloud penetration depth.

The objective of this paper is to demonstrate how passive solar radiance measurements can be used to identify possible inconsistencies of the ice crystal model used in radar-lidar retrievals. To this end, the paper is organized as follows: Section 2 briefly recapitulates the prerequisites needed for a successful combination of radar, lidar, and passive radiance measurements and will introduce the approach to validate radar-lidar retrieval results by radiative closure. The instruments and numerical methods used for this radiative closure study are introduced in Sec. 2.2 and Sec. 2.3. Section 3 then applies the presented approach to simultaneous radar, lidar, and passive radiance measurements from an airborne platform. Concluding, a case with unsuccessful radiative closure is analyzed and discussed in Sec. 4 using collocated in-situ measurements.

## 2 Methods

The following section introduces the methods used in the synergistic retrieval and its radiative closure study. It will also highlight the challenges and prerequisites for a successful retrieval of ice cloud microphysics from the combination of all three instruments. The prerequisites to reconcile the knowledge gained from radar, lidar, and passive radiance measurements are the following:

- The first prerequisite is simultaneous radar, lidar, and radiance measurements on a single platform. A temporal offset of minutes or a spatial offset larger than $1\,\mathrm{km}$ leads to errors for which a synergistic retrieval of ice cloud properties can no longer be trusted (Illingworth et al., 2000).

- Secondly, sufficiently realistic forward models are an essential building block of every retrieval. Without a consistent translation of cloud microphysical properties into signals of all three instruments, the retrieval can exhibit substantial biases. Scattering and absorption, as well as multiple scattering, should be described with as much complexity as necessary, while the models should remain as simple, and thus fast, as possible.

- Finally, the model which simplifies the variability of ice cloud microphysics and translates them into optical properties should be consistent among all three instruments. Different assumptions about the ice crystal shape or physically inconsistent particle properties would cause further biases which are inherently embedded in assumptions.

Fig. 1 illustrates our approach to obtain consistent microphysical, optical, and radiative properties of individual ice clouds as these prerequisites are met. Specifically, this study uses lidar (WALES) and radar (MIRA) measurements to retrieve the ice water content (IWC) and the ice crystal effective radius ($r_{\mathrm{eff}}$) using an optimal estimation framework (VarCloud). To check the retrieved microphysics for consistency, solar radiation reflected from these clouds is then forward simulated using





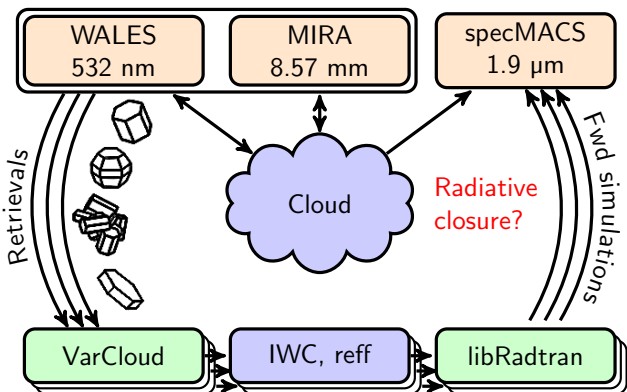

**Figure 1.** Overall strategy to validate the lidar-radar (WALES/MIRA) retrieval results (VarCloud) for different assumptions about the ice crystal shape by radiative closure between measured (specMACS) and simulated (libRadtran) solar radiances.

a sophisticated radiative transfer code (libRadtran) and compared against solar radiances measured by a spectroradiometer (specMACS) on the same platform. This is done multiple times using different assumptions about the ice crystal habit until radiative closure is achieved. The following subsection introduces the different instruments and methods in more detail.

## 2.1 Field campaign NAWDEX

During the North Atlantic Waveguide and Downstream Impact Experiment (NAWDEX; Schäfler et al., 2018), multiple research aircraft were deployed over the North Atlantic and western Europe in September and October 2016. The campaign was focused on the multi-scale observation of weather patterns associated with forecast errors of high impact weather over Europe. Here, a special focus was placed on rapidly intensifying cyclones and their associated warm conveyor belts (WCBs). For the duration of the campaign, multiple research aircraft were deployed for coordinated measurement flights: the German research aircraft HALO (High Altitude Long range, Krautstrunk and Giez, 2012), a modified Gulfstream G550 jet, and the SAFIRE French Falcon 20 operated from Iceland. For joint measurement flights, the BAe-146 research aircraft of the Facility for Airborne Atmospheric Measurements (FAAM, http://www.faam.ac.uk) operated from the United Kingdom.

## 2.2 Instruments

The lidar, radar and radiometer used in this study are part of the remote sensing payload of HALO. During various flight campaigns (NARVAL, NAWDEX, EUREC[4]A), the radar and lidar were deployed in the belly pod of HALO while the spectroradiometer was installed in the tail of the airplane (Fig. 2).





### 2.2.1 WALES

The DLR airborne lidar system WALES (Water vapor Lidar Experiment in Space) was built as a demonstrator for an ESA
proposed lidar mission in space to measure water vapor (Wirth et al., 2009). The WALES system has the capability for high
spectral resolution lidar (HSRL) measurements at 532 nm and for lidar depolarization measurements at 532 nm and 1064 nm.
Additionally, it measures water vapor mixing ratios from water vapor absorption bands around 935 nm (DIAL). In 2010, the
WALES system flew for the first time on the HALO aircraft and showed its potential for cirrus cloud and water vapour studies
(Groß et al., 2014).

### 2.2.2 HAMP MIRA

The HAMP MIRA instrument is a METEK Ka-band (35 GHz) cloud radar which can also determine the vertical velocity and
the depolarization of cloud particles. As part of the HALO microwave package (HAMP) it is deployed in the belly pod of
HALO. The instrument is well characterized and calibrated and proved to be in good agreement ($\pm 1$ dB) with the 94 GHz
cloud radars on board the French Falcon 20 aircraft and CloudSat during common flights (Ewald et al., 2019a).

### 2.2.3 specMACS

The specMACS imager is a combination of two imaging spectroradiometer in the visible to near infra-red (400–1000 nm)
and near infra-red (1000–2500 nm) wavelength regions. It measures spectral radiance with a spectral resolution of 3 nm in the
visible and 10 nm in the infra-red. As a push broom scanner, its spatial resolution is in the order of 10 m for cloud objects at
a distance of about 10 km The system is well characterized and calibrated (Ewald et al., 2015), while first retrievals for cloud
optical properties were developed (Zinner et al., 2016; Ewald et al., 2019b).

### 2.2.4 In-Situ measurements

For one of the flights (Sec. 3.2), simultaneous in-situ measurements of ice water content and ice particle size distributions
were made on board the FAAM BAe-146. During this flight (B984), the aircraft was equipped with a deep-cone Nevzorov
hot-wire probe (Korolev et al., 2013) which provides measurements of the bulk total and liquid water content. To enhance
the sensitivity for low ice water contents, the hot-wire measurements were corrected using the baseline correction proposed by
Abel et al. (2014). For flight B984, the BAe-146 was also equipped with the cloud imaging probes DMT CIP-15 and DMT CIP-
(Baumgardner et al., 2011) to measure the particle size distribution (PSD) of hydrometeors in 1 second intervals. For this
study, both instruments were fitted with deflection tips to reduce large ice crystal shattering which otherwise would contaminate
small particle number concentrations (Korolev et al., 2011). A detailed description of the cloud imaging instrumentation and
the processing of the data is given in Cotton et al. (2013). With a resolution of 15 μm, the CIP-15 probe covered the diameter
range $15 - 930$ μm of smaller cloud particles, while the CIP-100 probe sampled larger cloud particles with diameters between
$100 - 6200$ μm with a resolution of 100 μm. To obtain particle size distributions for the whole size range, the PSDs measured
by the CIP-15 probe were used up to a diameter of 700 μm and combined with PSDs measured by the CIP-100 probe above





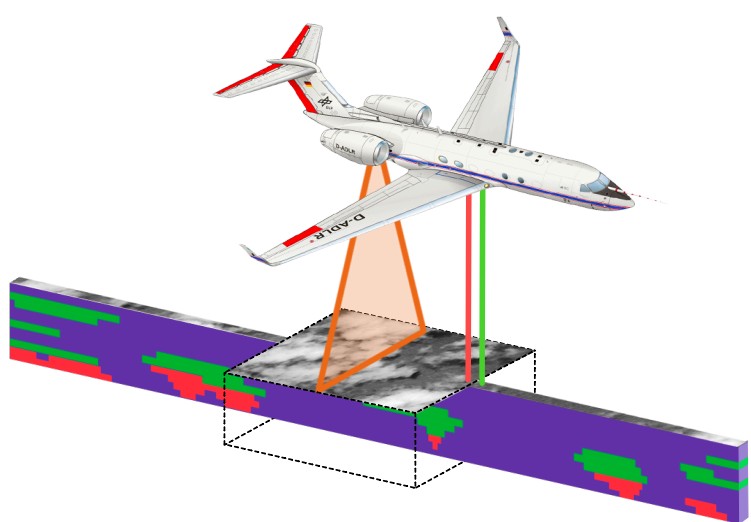

**Figure 2.** Combined lidar (WALES), radar (HAMP MIRA) and solar radiance measurements (specMACS) from the German High Altitude Long range (HALO) research aircraft. Lidar (green) and radar (red) provide along-track cross-sections through the atmosphere while the swath of the imager (orange) captures the across-track dimension of the scene.

that diameter. Due to the small sampling volume of the cloud imaging probes, the PSDs were furthermore averaged over 10
170 second intervals. These composite PSDs were then used to calculate ice crystal number concentrations for the whole diameter range.

### 2.3 Numerical methods

#### 2.3.1 Synergistic radar-lidar retrieval

The retrieval approach for the radar and lidar instruments is based on a variational optimal estimation algorithm (VarCloud;
175 Delanoë and Hogan, 2008), which combines radar, lidar, and thermal radiances in a unified framework. The retrieval is the basis of the *DARDAR Cloud* microphysics product for ice clouds on A-Train data (Delanoe and Hogan, 2010). The unique characteristic of this approach is its rigorous application of an online forward model developed by Hogan (2008) to simulate the multiple scattered lidar signal. It then uses the Jacobians from this forward model to update an a-priori microphysical profile to achieve convergence with the actual measurements. For this study, the most current retrieval version with updated ice cloud
180 microphysics of Cazenave et al. (2019) was used. The algorithm performs retrievals of extinction $\alpha$, IWC and $r_{\text{eff}}$. In addition, ice crystal number concentrations (ICNC) are derived from the microphysical best estimate. This method (*DARDAR Nice*) is described and has been thoroughly evaluated by Sourdeval et al. (2018) against a large amount of in-situ measurements. For this study, the VarCloud framework was adapted to the HALO instrumentation. To that end, the reflectivity lookup tables were extended to 35 GHz to include the wavelength of the cloud radar HAMP MIRA (see Sec. 2.3.2), while the wavelength ($532\,\mu m$)
185 and beam divergence of WALES was used in the lidar forward model.



### 2.3.2 Microphysical parameterization

The ice particle model and its scattering properties receive a detailed consideration as a central object of this study. A major challenge here is the different scattering regimes that need to be considered to explain radar, lidar, and passive radiance measurements. While the relationship between the mass and size of ice crystals is profoundly important for the backscatter of radar waves at millimeter wavelengths (Ham et al., 2017), their geometric cross-section has a decisive influence on lidar and passive solar radiance measurements (Holz et al., 2016). Even the shape of ice crystals influences the solar radiance reflected from ice clouds due to differences in the scattering phase function (Eichler et al., 2009).

A common model to simplify the variability of naturally occurring ice cloud particles is the effective ice particle density $\rho_{i,eff}$. It is defined as the ratio between the ice particle mass $M$ and the volume of a sphere that encloses the maximum diameter $D_{max}$ of the ice particle (Cotton et al., 2013). A frequent observation in in-situ measurements is the decreasing effective density of ice crystal as their maximum diameter $D_{max}$ increases (Brown and Francis, 1995; Cotton et al., 2013). Based on these measurements, the relationship between $D_{max}$ and $M$ is commonly described by a power law: $M(D_{max}) = a D_{max}^b$ (Mitchell et al., 1996; Heymsfield et al., 2010). For this study, the most recent M–D relationship for VarCloud with $a = 0.007$ and $b = 2.2$ was used (Cazenave et al., 2019). The M–D relationship also allows the calculation of the equivalent melted diameter $D_{eq}$ for a given $D_{max}$. Analogous, in-situ data was used by Heymsfield et al. (2013) to derive an A–D relationship to connect $D_{max}$ with the geometric cross-section $A$ of ice particles.

To describe the average scattering properties of ice particles, VarCloud uses the approximation by Hogan et al. (2012) of horizontally aligned oblate spheroids. This approximation simplifies the arbitrarily complex shape of ice particles with oblate spheroids with an aspect ratio of 0.6 while maintaining the maximum diameter $D_{max}$ and the total ice mass $M$. The dielectric properties of these *soft spheroids* with an effective density according to the M–D relationship are modeled as a blend of ice and air (Petty and Huang, 2010) using the effective medium approximation by Maxwell Garnett (1904). The radar cross-section $\sigma_{bck}$ is obtained by the T-matrix method of Mishchenko et al. (2004). The A–D relationship is used to calculate the visible extinction cross-section $\sigma_{ext} = 2 A(D)$ to be twice its geometric cross-section $A$ following the geometric optics limit here. The optical single-scattering properties of these spheroids, such as scattering phase function and asymmetry parameter $g$, are calculated using the T-matrix method.

The second ice crystal model tested in this study is the randomly oriented ice crystals described by Yang et al. (2000) with specific geometric shapes. The following study considers three ice crystal shapes, called habits: *solid columns*, *aggregates*, and *plates*. For reasons of consistency, the radar backscatter cross-section $\sigma_{bck}$ is calculated in the same way as for the *soft spheroids* using the corresponding M–D and A–D relationships given in Yang et al. (2000). For their optical properties, the well-established single scattering library of Yang et al. (2013) is used. In this library, the discrete dipole approximation, the T-matrix method, and an improved geometric optics method are combined to describe the more complex scattering of light by ice crystals with specific shapes.

To represent the variability of ice particle sizes within a cloud volume, a realistic and well-established particle size distribution (PSD) is used. Since PSDs are known to be highly variable (Intrieri et al., 1993), we choose the normalized PSD approach





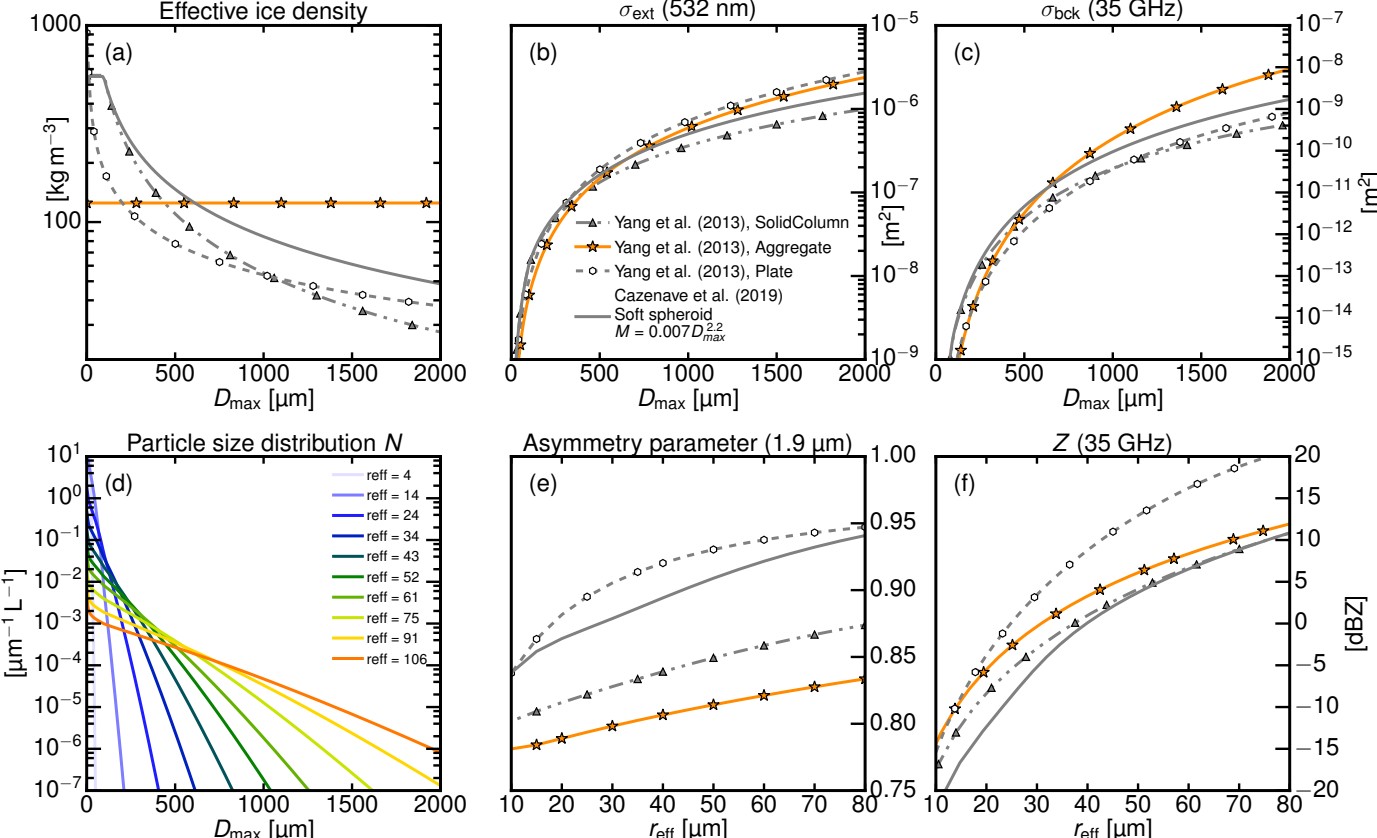

**Figure 3.** Microphysical, single scattering and bulk radiative properties of the different ice crystal models used in this study (gray line: *soft spheroid* approximation following Cazenave et al. (2019); symbol lines: specific ice crystal shapes following Yang et al., 2000). **(a)** Relationship between maximum dimension $D_{\max}$ and effective ice density for single ice crystals in $kg\,m^{-3}$, **(b)** extinction cross-section $\sigma_{\mathrm{ext}}$ at $532\,\mathrm{nm}$ and **(c)** radar backscatter cross-section $\sigma_{\mathrm{bck}}$ in $m^2$. **(d)** Particle size distributions of (Cazenave et al., 2019) for different effective radii and corresponding **(e)** asymmetry parameter at $1.9\,\mu m$ and radar reflectivity $Z$ at $35\,\mathrm{GHz}$ for an ice cloud with constant $\mathrm{IWC} = 1\,\mathrm{g\,m^{-3}}$.

by Delanoë et al. (2005) which is based on an extensive database of airborne in-situ measurements with updated parameters $\alpha_F = -0.262$ and $\beta_F = 1.754$ from Cazenave et al. (2019). The visible extinction $\alpha_v$ and the radar reflectivity $Z$ are then derived by integrating $\sigma_{\mathrm{ext}}$ and the radar backscatter cross-section $\sigma_{\mathrm{bck}}$ over this PSD:

$$\alpha_v = \qquad 2 \int N(D)\,A(D)\,dD \tag{1}$$

$$Z = \frac{\lambda^4}{|K|^2\,\pi^5} \int N(D)\,\sigma_{\mathrm{bck}}(D)\,dD \tag{2}$$





The same integration is done for the ice crystal mass $M(D)$ to obtain the corresponding IWC:

$$\text{IWC} = \int N(D) M(D) \, dD \tag{3}$$

Following Delanoë et al. (2014), the effective radius $r_\text{e}$ is calculated from $\alpha_v$ and IWC using the approximation of Foot (1988):

$$r_\text{e} = \frac{3}{2} \frac{\text{IWC}}{\rho_\text{ice} \alpha_v} \tag{4}$$

where $\rho_\text{ice} = 917\,\text{kg}\,\text{m}^{-3}$ is the density of ice.

Figure 3 summarizes the microphysical, single scattering, and bulk radiative properties for the *soft spheroid* approximation (gray line) used in Cazenave et al. (2019) and the specific ice crystal shapes (symbol line) of Yang et al. (2000). The upper panels in Fig. 3 show single particle properties as a function of the maximum dimension $D_\text{max}$, such as the effective ice density (Fig. 3a), the extinction cross-section $\sigma_\text{ext}$ at $532\,\text{nm}$ (Fig. 3b) and the radar backscatter cross-section $\sigma_\text{bck}$ in $m^2$

(Fig. 3c). For $D_\text{max} < 500\,\mu\text{m}$, Fig. 3a confirms that the specific ice crystal shapes (in particular *plates*) are less dense than the *soft spheroids* of Cazenave et al. (2019). Only larger *aggregates* ($D_\text{max} > 500\,\mu\text{m}$) have a higher effective density. The mostly two-dimensional *plates* have the largest extinction cross-section (Fig. 3b) in relation to $D_\text{max}$, followed by the complex structured *aggregates*, the *soft spheroids*, and the more needle-like *solid columns*. A similar behavior can be observed for $Z$, where *aggregates* and *solid columns* scatter less than *plates* when they have the same effective radius $r_\text{eff}$. Below $r_\text{eff} < 30\,\mu\text{m}$,

spheroids of same $r_\text{eff}$ show smaller $Z$ than *aggregates*, for $r_\text{eff} > 30\,\mu\text{m}$, spheroids are halfway between *solid columns* and *plates*.

### 2.3.3 Solar radiance forward modeling

At first, the missing vertical resolution of solar radiance measurements is a big disadvantage for this validation study. While VarCloud only retrieves properties of ice clouds, solar radiation can be reflected by all atmospheric constituents. As a conse-

quence, the radiance measurements can contain a mixture of information from ice clouds, underlying water clouds, aerosols, and the surface.

**Radiative transfer model**

In this study, the *DISORT* (Stamnes et al., 1988) solver was used to explore radiative transfer effects in one-dimensional, multilayer cloud scenes. For cloud scenes reconstructed from HALO measurements, more realistic forward simulations of

reflected solar radiation were done using the *Monte Carlo code for the physically correct tracing of photons in cloudy atmospheres* (*MYSTIC*; Mayer, 2009). Both models are part of the radiative transfer library *libRadtran* (Mayer and Kylling, 2005; Emde et al., 2016) which also includes the single-scattering properties of Yang et al. (2013). Atmospheric absorption is considered using the representative wavelengths absorption parametrization (*REPTRAN*; Gasteiger et al., 2014) which is based on the HITRAN absorption database (Rothman et al., 2005). As shown by Zinner et al. (2019), the medium resolution ($\text{cm}^{-1}$)

of REPTRAN is sufficient to model the spectral resolution of specMACS after convolving it with its spectral response (e.g.





$\Delta\lambda = 6.4\,\text{nm}$ at $1900\,\text{nm}$, Ewald et al., 2016). For the following sensitivity study, the standard summer mid-latitude profiles by Anderson et al. (1986) were used.

**Exclusion of surface and water cloud reflection**

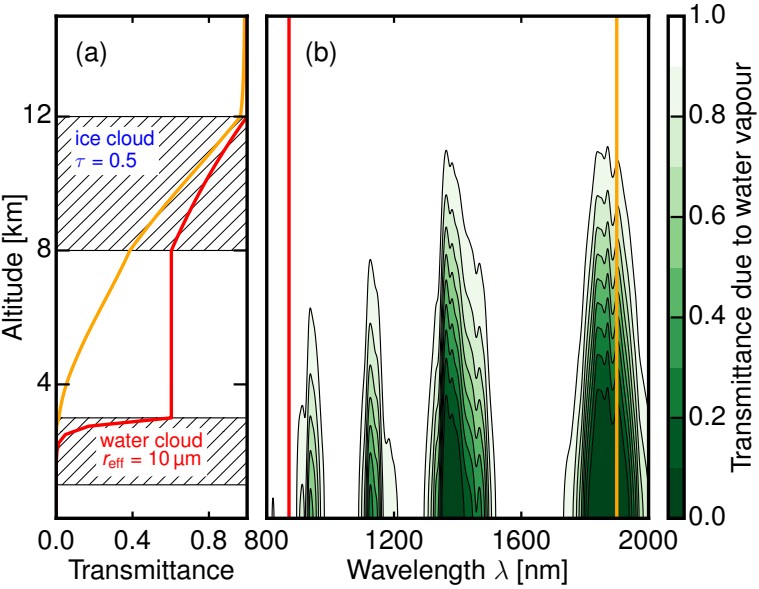

**Figure 4. (a)** Atmospheric transmittance of a water cloud and overlying ice cloud layer at $870\,\text{nm}$ (red line) and $1.9\,\mu\text{m}$ (orange line) **(b)** The spectral transmittance of atmospheric water vapour acting as a vertical weighting function.

To overcome the previously mentioned problem of multi-layer scenes for passive remote sensing, Gao et al. (1993) suggested

exploiting the water vapor absorption band at $1.38\,\mu\text{m}$ to detect thin cirrus clouds with the Airborne Visible/Infrared Imaging Spectrometer (AVIRIS). The technique takes advantage of the fact that cirrus clouds and large parts of other ice clouds are mostly located above the atmospheric water vapour column. In a strong water vapor absorption band, a downward-looking sensor flying above $10\,\text{km}$ receives almost no solar radiation scattered from the surface or low-level clouds. In contrast, the solar radiation scattered by high level clouds stands out above this black and homogeneous background. This technique is also

used to monitor the reflectance (Gao and Kaufman, 1994) and to retrieve the optical thickness (Meyer and Platnick, 2010) of cirrus clouds globally using the Moderate Resolution Imaging Spectrometer (MODIS).

With specMACS, all water vapour absorption bands up to $2.5\,\mu\text{m}$ in the near-infrared wavelength region are readily available. Figure 4 explores and illustrates the technique to exclude the contribution of the surface and low-level water clouds in multilayer scenes observed with specMACS. In this experiment, a water cloud layer with a fixed effective radius $r_{\text{eff,w}}$ of $10\,\mu\text{m}$ was

superimposed with an ice cloud layer with a fixed optical thickness $\tau_i$ of 0.5. Subsequently, *DISORT* was used to calculate the spectral transmittance of that cloud scene for solar radiation between $800\,\text{nm}$ and $2.5\,\mu\text{m}$. Figure 4a shows the atmospheric transmittance at $870\,\text{nm}$ (red line) and $1.9\,\mu\text{m}$ (orange line) as a function of altitude. It is evident how the atmosphere is



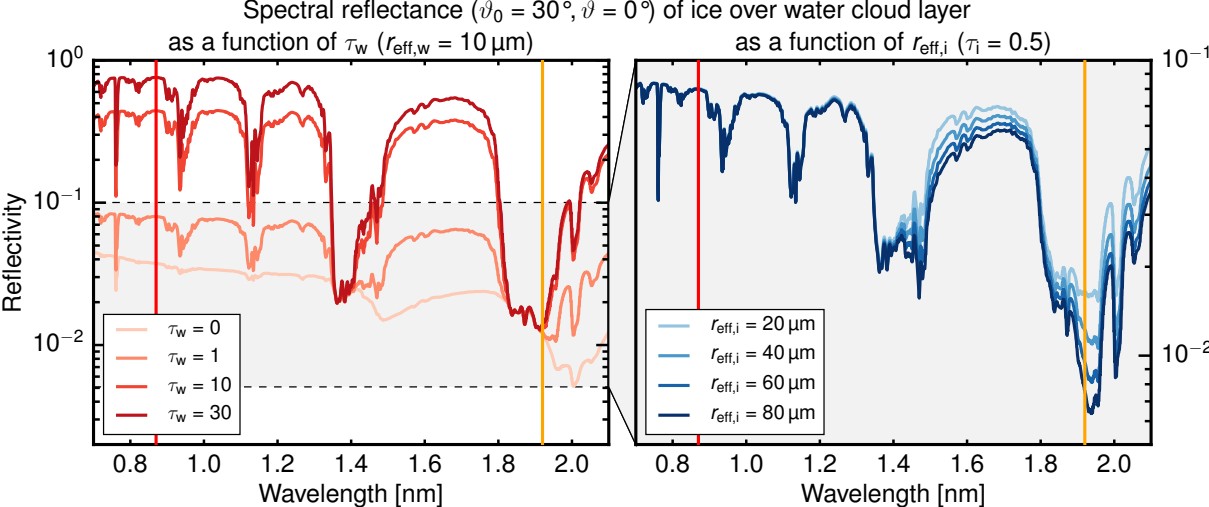

**Figure 5.** Spectral reflectance of an ice over water cloud layer as sketched in Fig. 4a for the nadir ($\vartheta = 0°$) perspective and a solar zenith angle of $\vartheta_0 = 30°$. **(left)** Results (red lines) for varying optical thickness $\tau_w$ of the water cloud layer and **(right)** results (blue lines) for varying ice crystal size $r_{\text{eff,i}}$ of the ice cloud layer.

semi-transparent down to the water cloud layer in a so-called *window channel* at $870\,\text{nm}$ and how absorption by water vapour confines the solar radiation at $1.9\,\mu\text{m}$ to the upper troposphere. Fig. 4b illustrates how the spectral transmittance of atmospheric

water vapour acts as a vertical weighting function for reflected photons. The most opaque water vapor bands are centered at $1.38\,\mu\text{m}$ and $1.9\,\mu\text{m}$ within the wavelength range accessible with specMACS.

While the more commonly used *cirrus band* at $1.38\,\mu\text{m}$ is almost as opaque as the band at $1.9\,\mu\text{m}$, the latter has a significant advantage for the radiative closure study: the absorption coefficient of ice exhibits a much stronger maximum close to $1.9\,\mu\text{m}$ which gives this channel a sensitivity to ice crystal size. To analyze this unique combination of sensitivity and opaqueness,

the spectral reflectance of this scene was calculated while varying the ice crystal size $r_{\text{eff,i}}$ in the cirrus layer and the optical thickness $\tau_w$ of the underlying water cloud layer. Figure 5 (left) shows the results for different $\tau_w$ ($r_{\text{eff,w}} = 10\,\mu\text{m}$) and fixed optical thickness $\tau_i$ of 0.5. While the reflectance at $870\,\text{nm}$ increases from 0.03 to 0.7 as $\tau_w$ increases from 0 to 30, it remains invariant of $\tau_w$ at both water vapour absorption bands ($1.9\,\mu\text{m}$ as well as $1.38\,\mu\text{m}$. When the ice crystal size $r_{\text{eff,i}}$ is modified (Fig. 5, right), however, the spectral reflectance shows a different characteristic. While the reflectance is cut in half (0.016 to

0.007) as ice crystal size increases from $r_{\text{eff,i}} = 40\,\mu\text{m}$ to $r_{\text{eff,i}} = 80\,\mu\text{m}$ at $1.9\,\mu\text{m}$, no large variation can be observed for $1.38\,\mu\text{m}$. The sensitivity for $r_{\text{eff,i}}$ appears at slightly larger wavelengths ($1.4\,\mu\text{m}$) for which the atmosphere becomes transparent down to the water cloud layer again. Hence, the $1.9\,\mu\text{m}$ water vapour absorption band is the only sufficiently opaque wavelength region accessible with specMACS which simultaneously shows a sensitivity to ice crystal size.





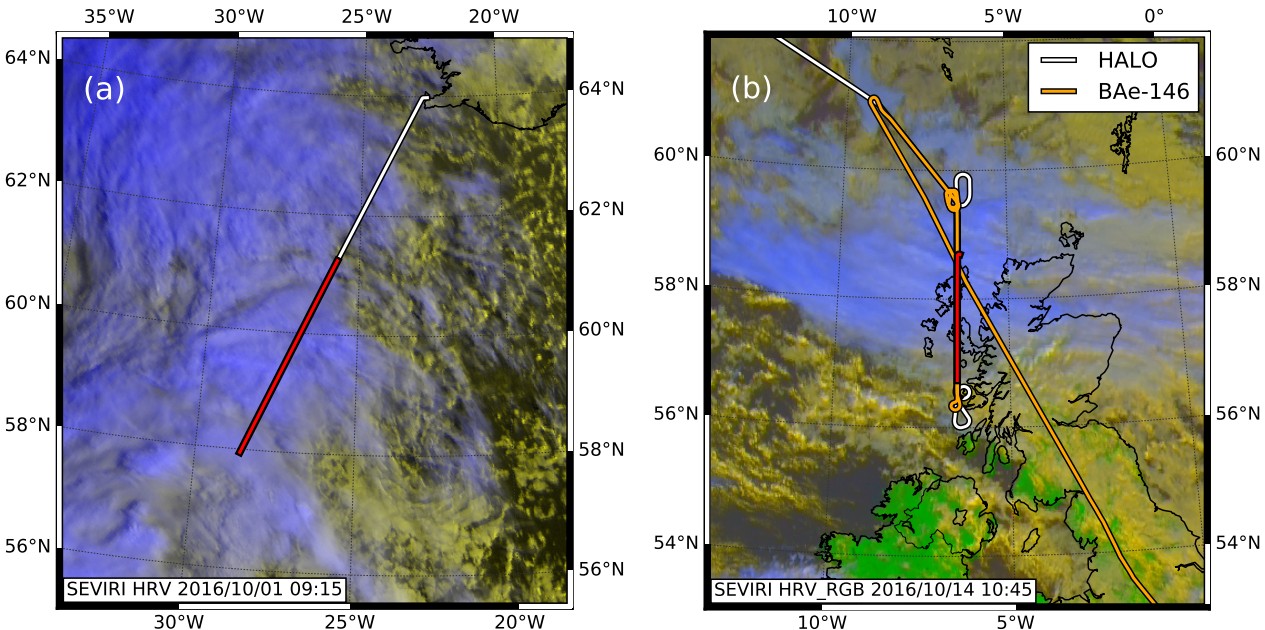

**Figure 6. (a)** SEVIRI satellite image of the case discussed in Fig. 7 (red section) where HALO (white) measured the Cirrus outflow of a WCB on 1 October 2016 in a region south of Iceland. **(b)** SEVIRI satellite image of the case discussed in Fig. 8. On 14 October 2016, the HALO (white) and the FAAM BAe-146 (orange) research aircraft flew a coordinated flight leg (red section) over ice clouds within an occluded front west of the Scottish Highlands. Copyright 2020 EUMETSAT.

## 3 Solar radiance closure study

### 3.1 Case 1: Cirrus outflow of a WCB

The first case study was measured during the 6[th] research flight (RF06) of HALO on 1 October 2016. The scientific objective of the flight was a rapidly intensifying cyclone south-west of Iceland, named the Stalactite cyclone due to its stalactite-like tropopause trough (Schäfler et al., 2018). Its rapid development occurred between 29 September and 2 October in the context of a large-scale upper-level trough over Greenland. On 1 October, its center was located at about $50° N, 35° W$ with an intense WCB located in the upstream region of a warm subtropical air mass. The strong ascent led to a strong ridge building over Iceland and the subsequent formation of a Scandinavian blocking situation (Maddison et al., 2019). A satellite image in Fig. 6a reveals the flight path (white) and the flight leg (red section) considered in this case study. The panels in Fig. 7 show measurements and retrieved ice microphysics that where made between $08 : 55 - 09 : 25$ UTC above a cirrus cloud layer at the eastern flank of the upper-level divergent outflow of the WCB. Between $61.2° N, 25.8° W$ and $57.9° N, 28.6° W$, this cirrus cloud deck appeared above a shallow marine cloud deck and deepened during the flight leg towards the center of the cyclone.

The top-down perspective along the flight path is given in Fig. 7a by a true-color image which was acquired by specMACS. The corresponding vertical perspective obtained by the active remote sensing instruments is shown in Fig. 7b with the attenu-



ated backscatter coefficient measured by WALES at $532\,\mathrm{nm}$ and in Fig. 7c with the equivalent effective reflectivity $Z_e$ measured by HAMP MIRA at $35\,\mathrm{GHz}$. Fig. 7b and Fig. 7c illustrate the complementary nature of radar and lidar measurements: while

the lidar can contribute detailed structures in optically thin layers on cloud top, the cloud radar retrieves signals from deep within the cloud where the lidar signal is already extinguished. This synergy is used to retrieve IWC and $r_\mathrm{e}$ using the VarCloud framework described in Sec. 2.3.1. Figure 7d and Fig. 7e show the retrieved IWC and the retrieved ice crystal effective radius using the microphysical parameterization of Cazenave et al. (2019) in VarCloud. While ice crystals are very small at cloud top ($r_\mathrm{e} = 20\,\mathrm{\mu m}$), their size increases considerably while sedimenting downward to reach $r_\mathrm{e} = 80\,\mathrm{\mu m}$ at the bottom of the cirrus

layer.

## 3.2    Case 2: Occluded front clouds

The second case study was measured during the 11[th] research flight (RF11) of HALO on 14 October 2016. The scientific objective was the collocated measurement of a frontal cloud system with three aircraft and a joint underpass of the CALIPSO/CloudSat satellite constellation to characterize and validate synergies obtained from radar, lidar, and radiometer

measurements. The frontal cloud system was located over Scotland and was associated with a cut-off low just west of Ireland. On the leading edge of this low, a moist and warm air mass was advected northward over the North Sea and lifted to form an occluded front. Over the day, the front remained almost stationary with a southeastern flow over the Scottish Highlands.

Over the sea between the Scottish Highlands and the Outer Hebrides, HALO, the SAFIRE Falcon and the FAAM BAe-146 performed a common flight leg staggered at different altitudes above this occluded front. The satellite image in Fig. 6b gives

an overview of the cloud scene, the HALO (white) and FAAM BAe-146 (orange) flight track and the common flight leg (red section). While HALO flew over the cloud layer at an altitude of $13.5\,\mathrm{km}$, the FAAM BAe-146 performed a profiling flight pattern within the radar-lidar curtain. Fig. 8 shows the measurements made on HALO between $10:30 - 10:52$ while all three aircraft flew a south-north cross-section over the occluded front along $6.5°\,W$ longitude between $58.1°\,N$ and $59.4°\,N$. Fig. 8a shows again a true-color image measured with specMACS for a zoomed section between $10:30 - 10:33$ along the flight path.

The attenuated backscatter coefficient in Fig. 8b shows very strong backscatter peaks embedded within multiple cloud decks at an altitude of $5\,\mathrm{km}$ which rise stepwise to a continuous cloud deck at an altitude of $8\,\mathrm{km}$ in the second part of the cross-section. Ahead and trailing the front, multiple supercooled cloud layers can be identified by their strong backscatter and attenuation. Overall, the lidar signal is much extinguished more rapidly compared to the case shown in Fig. 7b. The equivalent effective reflectivity $Z_e$ in Fig. 8 shows a deep ice cloud layer with precipitation to the ground and mixed-phase regions above a melting

layer at $1.5\,\mathrm{km}$ altitude. The overlap of radar and lidar measurements is smaller in contrast to the first case (Sec. 3.1). To exclude obvious mixed-phase regions, the VarCloud retrieval was only applied to measurements with air temperatures below $-15°\,\mathrm{C}$ and down to $4\,\mathrm{km}$ altitude. Like before, the last two panels (Figure 8d and Figure 8e) present the retrieved IWC and the retrieved effective radius for the default microphysical parameterization of Cazenave et al. (2019).

**Figure 7.** Remote sensing of a cirrus layer measured with HALO on 1 October 2016 during the NAWDEX campaign. **(a)** True-color image acquired by the hyperspectral cloud imager specMACS (Ewald et al., 2016) along the flight path, **(b)** attenuated backscatter coefficient measured with the WALES lidar at 532 nm and corresponding **(c)** equivalent effective reflectivity $Z_e$ measured with the cloud radar HAMP MIRA at 35 GHz. **(d)** Ice water content and **(e)** effective radius of ice crystals retrieved by combining information from lidar (Fig. 7b) and (Fig. 7c) radar using the VarCloud framework.

**Figure 8.** Remote sensing of a cloud layer measured with HALO on 14 October 2016 during the NAWDEX campaign. **(a)** Spectral radiance at $1.9\,\mu m$ acquired by specMACS along the flight path, **(b)** and **(c)** same as Fig. 7. **(d)** Ice water content and **(e)** effective radius retrieved by VarCloud. As an overlay in Panel d), in-situ measured IWCs are plotted along the BAa-146 flight path (drawn line) with the spatial region (dashed lines) considered for the in-situ comparison in Fig. 10.





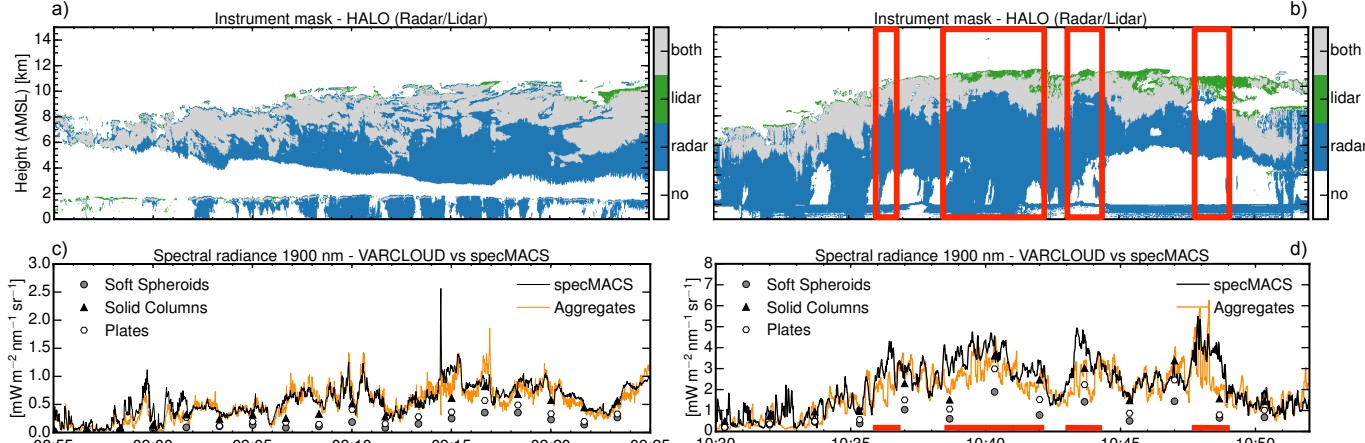

**Figure 9.** Radiative closure study for the measurements shown in Fig. 7 and Fig. 8. **(a, b)** Instrument masks indicating regions with measurements from lidar-only, radar-only and both instruments. The overlap region for which radar and lidar measurement are available is much larger for the first case. **(c, d)** Forward modeled solar radiances (orange lines) compare well with measured solar radiances (black lines) for the case with large instrument overlap **(a)** but disagree for the case with small overlap region **(b)** when *Aggregates* are used. *Soft Spheroids* (gray circles), *solid columns* (black triangles) and *plates* (white hexagons) lead to an underestimation of reflected solar radiation in both cases.

## 3.3 Comparison with measured radiances

For both cases discussed in the previous Sections 3.1 and 3.2, VarCloud was applied using the various microphysical assumptions described in Sect. 2.3.2: once using the default parameterization of Cazenave et al. (2019) and furthermore with the M–D and A–D relationships for the specific ice crystal habits of Yang et al. (2000). The retrieved IWC and $r_e$ were then used as input cloud fields to simulate the reflected solar radiation at $1.9\,\mu m$ using optical properties corresponding to each microphysical parameterization as described in Sec. 2. Subsequently, the simulated solar radiances were compared with real measurements obtained with specMACS.

Figure 9c shows the comparison of measured and simulated solar radiances for RF06 on 1 October 2016. The relative variation of reflected radiance can be reproduced remarkably well by all microphysics tested. Over the whole scene, however, substantial biases become apparent. With their very strong forward scattering (see asymmetry parameter in Fig. 3e), *plates*, as well as *soft spheroids*, lead to a very strong underestimation of reflected solar radiation of $-51\%$ and $-71\%$ respectively. A step closer to radiative closure can be achieved when ice crystals with less forward scattering are used. While *Solid columns* still lead to an underestimation of reflected solar radiation $(-22\%)$, the habit assumption with the smallest asymmetry parameter, *Aggregates*, can reproduce the measured solar radiances remarkably well $(-5\%)$.

For the second case introduced in Sec. 3.2, radiative closure turned out to be harder to achieve for all microphysics tested. Over the whole scene, the assumption of *plates* or *soft spheroids* leads to a similar strong underestimation of reflected solar radiation $(-50\%$ or $-69\%$, respectively) like in the first case. The radiative closure for *solid columns* and *aggregates* with





an underestimation of $-30\%$ and $-17\%$, respectively, is now less convincing compared to the first case. While radiative closure could be achieved remarkably well for certain sections of the flight (e.g. $10:44 - 10:48$ UTC) using *aggregates*, a closer inspection reveals cloud regions responsible for the overall underperformance. The comparison of measured and simulated radiances in Figure 9d shows multiple regions where all used microphysics are unable to produce the higher spectral

radiances measured by specMACS. This is particularly obvious during the period between $10:38 - 10:42$, $10:43 - 10:44$ and $10:48 - 10:49$ UTC. Here, measured radiances are up to two times larger than the simulated radiances. These regions also coincide with layers of a very strong lidar backscatter at cloud top for which the lidar signal is quickly extinguished. This leads to a reduced overlap between lidar and radar measurements with negative consequences for the exploitation of synergies.

The overlap of radar and lidar are the gray areas in the instrument masks shown in Fig. 9a for RF06 and in Fig. 9b for

RF11. Here, the different vertical extent of the overlap region becomes apparent between both cases. When the overlap region is large (Fig. 9a), forward modeled solar radiances (using *aggregates*) compare well with measured solar radiances (Fig. 9c). In contrast, the radiative closure completely fails for cloud regions where the overlap region is small (marked by red regions in Fig. 9d). These regions are dominated by radar measurements and, in turn, have to rely heavily on assumptions of the ice crystal shape.

# 4    Comparison of in-situ with remote sensing

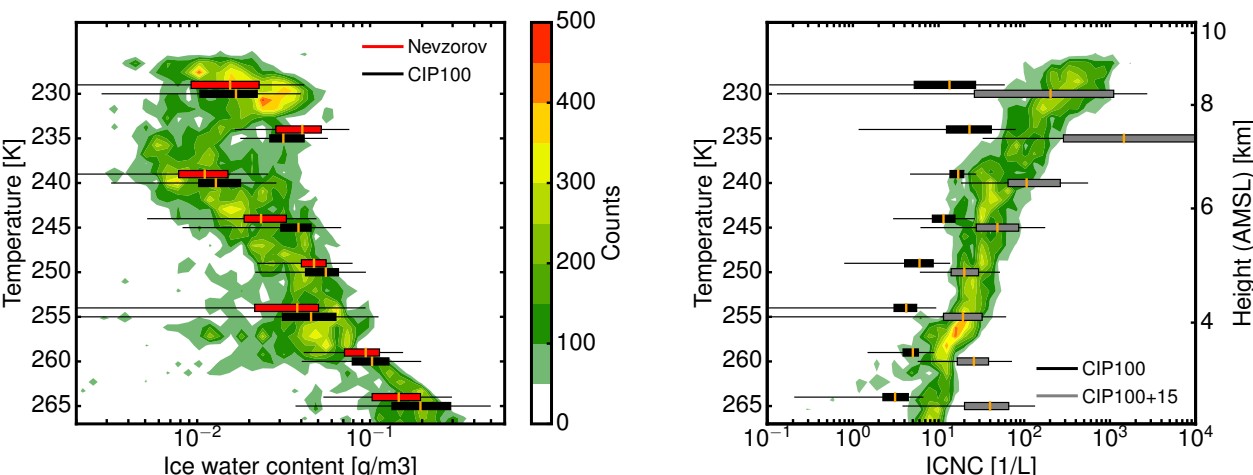

**Figure 10.** Comparison of VarCloud results derived from HALO measurements with in-situ measurements on board the BAe-146 for the joint flight leg. **(left)** Retrieved ice water content (contour) against Nevzorov hot-wire (red boxplot) and CIP-100 (black boxplot) probe measurements. **(right)** Retrieved ice crystal number concentration (contour) against the composite measurement of CIP-15 and CIP-100 (gray boxplot) and CIP-100 (black boxplot) alone.

Collocated in-situ measurements from the BAe-146 are available for the case study (shown in Fig. 8) with the partly failed radiative closure (shown in Fig. 9c). The in-situ data and its processing is described Sec 2.2.4. Figure 10 summarizes the





comparison of retrieved and measured profiles of ice cloud microphysics. Between $10:35$ and $11:00$, the BAe-146 sampled
in-situ data along the same measurement curtain in a stepwise descent from $8\,\mathrm{km}$ down to $2\,\mathrm{km}$. To ensure comparability, the
comparison with in-situ data is only performed for VarCloud results within a spatial vicinity of $\pm500\,\mathrm{m}$ of the BAe-146 flight
path. The temporal offset is limited to $15\,\mathrm{min}$, with a better temporal coincidence ($<5\,\mathrm{min}$) for the flyover of BAe-146 by
HALO between $8\,\mathrm{km}$ and $4.5\,\mathrm{km}$ altitude. Figure 8d shows IWCs retrieved by VarCloud superimposed with IWCs measured
along the BAe-146 flight path. Here, the spatial region considered for comparison is delimited by the dashed lines. For the
following study, the in-situ data was binned by temperature in steps of $5\,\mathrm{K}$ to obtain reliable statistics of the vertical profile.
The following comparison are in-cloud statistics, where retrieval and in-situ data with IWCs smaller than $10^{-3}\,\mathrm{gm^{-3}}$ have been
discarded.

In the following, IWCs retrieved with VarCloud are validated using data from the Nevzorov hot-wire as well as the CIP-100.
Figure 10 (left) shows box plots of the averaged IWC profile measured by the Nevzorov hot-wire (red) and the CIP-100 (black).
Here, the boxes show the lower and upper quartile of measured IWCs while the whiskers give the maximum and minimum
values found (excluding outliers outside the 1.5 interquartile range). The median IWC is shown by the orange vertical lines
through the boxes. The contour in the background of Figure 10 (left) represents the retrieved IWC using the assumptions of
Cazenave et al. (2019). While the overall observation of increasing IWC with increasing air temperature is reproduced well
by VarCloud, biases become apparent at cloud top and deeper within the cloud in comparison with the Nevzorov hot-wire
measurements. At cloud top, the median IWC is first sightly overestimated by VarCloud by $+10\,\%$ at $T=230\,\mathrm{K}$, but then
strongly underestimated by up to $-70\,\%$ at $T=235\,\mathrm{K}$. At around $T=240\,\mathrm{K}$ and below, the agreement with in situ IWCs is
remarkably good. Between $T=240\,\mathrm{K}$ and $T=255\,\mathrm{K}$, the median of the retrieved IWC is well inside the lower and upper
quantile of the in-situ data with a small negative bias of up to $-15\,\%$. At even lower altitudes and with air temperatures rising
to the melting point of ice, the retrieved IWC still agrees well with in-situ data with a slight overestimation of up to $+20\,\%$.
Throughout the whole profile, the hot-wire data is in line with the CIP-100 probe measurements, with a slight disagreement of
less than $25\,\%$ at $T=245\,\mathrm{K}$.

In the same manner, the retrieved and measured ICNCs are compared in Fig. 10 (right). This comparison is once done for the
composite PSDs from the CIP-15 and CIP-100 probe (gray boxplot) and once including only larger particles from the CIP-100
probe (black boxplot) to analyze the contribution of very small ice crystals to the ICNC. Here, the challenging situation just
below the top of the cloud layer is even more obvious. While the retrieval gets the ICNC almost right directly at cloud top
($230\,\mathrm{K}:280\,\mathrm{L^{-1}}$ vs. $200\,\mathrm{L^{-1}}$), it misses the extraordinary high ICNC slightly below ($235\,\mathrm{K}:130\,\mathrm{L^{-1}}$ vs. $1500\,\mathrm{L^{-1}}$). Below
this region and similar to the IWC validation, VarCloud agrees remarkably well with the ICNC of the composite PSD. The very
high values just below cloud top ($235\,\mathrm{K}$) can be mainly explained by a high number of very small particles when comparing
ICNCs from the combined CIP probes with ICNCs from the CIP-100 probe alone.



## 5 Discussion

In the first case study (Sec. 3.1), radiative closure could be achieved by changing the assumption of the ice crystal shape. While the standard *soft spheroid* approximation led to a strong underestimation of reflected solar radiation, radiative closure could be achieved when using *aggregates*. At wavelengths without strong absorption of light by ice, reflected solar radiation from ice clouds is mainly governed by the optical thickness and the scattering phase function of its particles (Fu and Takano, 1994). For cloud layers with the same optical thickness, ice crystal shapes with a stronger forward scattering (i.e. larger asymmetry parameter) led to lower reflected radiance at cloud top (Eichler et al., 2009). This is in line with the first case study, where the ice crystals with a large asymmetry parameter, like *plates* and *soft spheroids*, led to a strong underestimation of reflected solar radiation.

It is worth mentioning that the *soft spheroid* assumption led to the lowest radiances although *plates* of the same effective radius have a larger asymmetry parameter (see Fig. 3e). This apparent contradiction is resolved when the intermediate VarCloud results, in particular the retrieved effective radii, are compared between the ice crystal habits (Fig A1 and Fig. 7e). Here, VarCloud retrieves significantly smaller $r_{\mathrm{eff}}$ for the *plates* assumption. This can be explained with Fig. 3f, where the radar reflectivity $Z$ is shown as a function of $r_{\mathrm{eff}}$ for an ice cloud with constant $\mathrm{IWC} = 1\,\mathrm{g\,m^{-3}}$. For an observed value of $Z$, *plates* always have the smallest $r_{\mathrm{eff}}$. If one exchanges $Z$ with particle mass, this observation is in line with the definition of $r_{\mathrm{eff}}$ in Eq. 4. For the same particle mass and with $r_{\mathrm{eff}}$ defined as the ratio of particle mass and visible extinction, the primarily two-dimensional plates have the smallest $r_{\mathrm{eff}}$ since they have the largest visible extinction cross-section compared to the other habits. In turn, the *soft spheroid* assumption thus yields a larger $r_{\mathrm{eff}}$ and thus larger asymmetry parameter compared to the to *plate* assumption (see Fig. 3e). This explains the strongest underestimation of reflected solar radiation by *soft spheroids*, followed by *plates* and the better agreement for *solid columns* and *aggregates*.

In contrast, changing the assumption of the ice crystal shape could not explain all discrepancies found between the forward simulated and measured radiances for the second case (Sec. 3.2). This is an indication that there are further challenges beyond the ice crystal habit assumption for this cloud scene. The in-situ data suggests a very high ICNC with predominately small ice crystals which poses a problem on several levels: (1) Cloud regions with high ICNC and small ice crystals are barely visible in cloud radar measurements, while the lidar signal is quickly extinguished. This has a negative consequence on the instrument overlap which is needed to determine IWC and $r_{\mathrm{eff}}$ without relying too heavily on a-priori profiles. (2) Delanoë et al. (2014) and Cazenave et al. (2019) included particles down to a minimum diameter of $50\,\mathrm{\mu m}$ to fit the shape of the normalized PSD shape (Fig. 3d) to in-situ data corrected for ice shattering effects. However, the large spread of almost two magnitudes between the ICNC measured by the CIP-15 and CIP-100 probe is an indication that the normalized PSD can no longer capture the PSD shape of this specific cloud region at low temperatures. (3) Furthermore, there is a very distinct jump in ICNC between $240\,\mathrm{K}$ and $235\,\mathrm{K}$. However, cubic spline basis functions with a sampling distance of $240\,\mathrm{m}$ are used to smooth the microphysical profile of the ice crystal number concentration and to stabilize the performance of the VarCloud algorithm. Since VarCloud is designed to approximate the radar and lidar signals anyhow, this could lead to an undesired *buffering* into other microphysical variables, like the lidar ratio or extinction, in thin ice cloud layers.



## 6 Conclusions

This study demonstrated how passive solar radiance measurements can be used to test the well-established variational approach
VarCloud and to adapt the assumed ice crystal model to be consistent with radar-lidar as well as radiance measurements. While
active remote sensing is capable of providing vertical backscatter profiles, the inversion to ice cloud microphysics relies heavily
on the assumption of the prevalent ice particle shape and its mass-size relationship. On the basis of two airborne-measured case
studies, this paper analyzed VarCloud results for different ice crystal habit assumptions. The VarCloud results for the different
habit assumptions were then used to simulate reflected solar radiances. Through radiative closure with simultaneously measured
solar radiances, the performance of VarCloud could then be tested for the different habit assumptions. Besides the standard *soft
spheroid* approximation of VarCloud, three specific ice crystal habits (*solid columns*, *aggregates*, and *plates*) were tested for
their ability to reconcile radar, lidar, and solar radiance measurements. To ensure physical consistency this was done for the
radar-lidar retrieval, as well as for the forward simulations of solar radiance. To exclude the contribution of surface reflection
and solar radiation reflected by low-level liquid clouds, this radiative closure study was done at $\lambda = 1.9\,\mu m$. This technique
exploits the strong water vapor absorption, where mainly light reflected by cirrus and high-altitude ice clouds is contributing
to the measured radiance. At this wavelength, radiative closure could be achieved in on case study by changing the ice crystal
habit assumption from the *soft spheroid* model of Cazenave et al. (2019) (underestimation of solar radiation by $-71\,\%$) to
the *aggregate* model of Yang et al. (2000) (underestimation of solar radiation by $-5\,\%$). In a second case study, changing the
assumption of the ice crystal shape to *aggregates* led to an improved radiative closure, too. In contrast to the first case study,
this could not explain all discrepancies found for certain cloud sections between the forward simulated and measured radiances.
Here, collocated in-situ measurements revealed very high ICNCs slightly below cloud top which strongly reduced the overlap
of radar and lidar measurements.

In the light of these findings, the following conclusions can be drawn:

– In both cases and for all tested ice habit assumptions, the radar-lidar framework VarCloud found a microphysical state
which could explain the radar and lidar signals within their measurement uncertainties. Similar residuals between the
forward simulations and radar and lidar measurements did not allow to discriminate the best-fitting ice crystal habit for
the first case study (Sec. 3.1) nor did it indicate a problem for the second case study (Sec. 3.2).

– This is an expected behavior of an under-determined problem with two measurements ($\beta_a$ and $Z_e$) but three unknowns
(IWC, $r_{\text{eff}}$, ice habit). Here, an additional measurement using a completely different remote sensing technique, e.g.
passive remote sensing of reflected solar radiation, is an urgently needed benchmark to assess the quality of the radar-
lidar result and to identify inconsistencies of the used assumptions.

– In the case of a large radar-lidar overlap, and hence two measurements, the reflected solar radiation can help to narrow
down the ice crystal shape assumption. Here, the sensitivity to the asymmetry parameter of the scatterer in the reflected
solar radiation is key to obtain additional information about the ice crystal shape.





– At first glance, passive solar radiance falls short in comparison with the rich vertical insight of radar and lidar measure-
ments. A closer inspection reveals the unique strength of passive measurements being the product of an integral over
the cloud profile: While radar and lidar signals contain only information in the exact backscatter direction of the ice
crystals, reflected solar radiation is the product of a multiple scattering process and thus sensitive to the full scattering
phase function of the ice crystals.

Observations of reflected solar radiance thus complement the active profiling technique. In two case studies, this work could
show how the proposed radiative closure technique can be used to test and improve the performance of a radar-lidar retrieval:

1. The closure with measured radiances can help to obtain consistent cloud properties with correct radiative properties in
the solar spectrum. This is especially important for studies which are using radar-lidar retrieval results to assess the
radiative effect of ice clouds.

2. Radiative closure can furthermore be used to assess the performance of the radar-lidar technique and to identify regions
with unreliable retrieval results. In this study, the radiative closure technique was able to spot cloud regions with a very
high ice crystal number concentration and, in turn, unreliable VarCloud results which would have been otherwise missed.

While this study demonstrated the radiative closure technique for VarCloud, further studies are now required which are
beyond the scope of this manuscript:

– A further study should assess the VarCloud performance on the basis of a sound statistical data set using existing mea-
surements made during prior airborne campaigns.

– A method should be developed to incorporate the solar radiance measurements already during the VarCloud optimal
estimate. This should naturally lead to a better constrain of the ice crystal model and to a physically more consistent
retrieval result.

– Right now, VarCloud as well as this study, assume one ice crystal model (e.g. a fixed M–D relationship). Various studies
found a large variability of ice cloud among clouds in different geographical regions, as well as within individual clouds
volumes (Comstock et al., 2007; Deng et al., 2012; Xu and Mace, 2016). To that end, a further degree of freedom (e.g. a
parameter of the ice crystal model) has to be introduced which can be seamlessly changed throughout the microphysical
profile.

Recent years have brought significant progress towards an integrated approach to combine multiple remote sensing instruments.
In the context of the tenth anniversary of the two A-Train profilers CloudSat and CALIPSO and the upcoming launch of
EarthCARE, progress is due to harmonize existing radar-lidar retrieval techniques with passive measurements. In this context,
the seamless exploitation of passive solar radiances within VarCloud will be a next step towards a better understanding of ice
cloud microphysics.



*Data availability.* The data set of *HAMP MIRA* is available in ESSD with the identifier https://doi.org/10.5194/essd-2018-116. The data set of *WALES* is available via halo-db.pa.op.dlr.de. SEVIRI L1 data (HRIT) were provided by EUMETSAT via EUMETCast. The in-situ and *specMACS* data are provided upon request.

## Appendix A: Influence of ice crystal habit on $r_{eff}$.

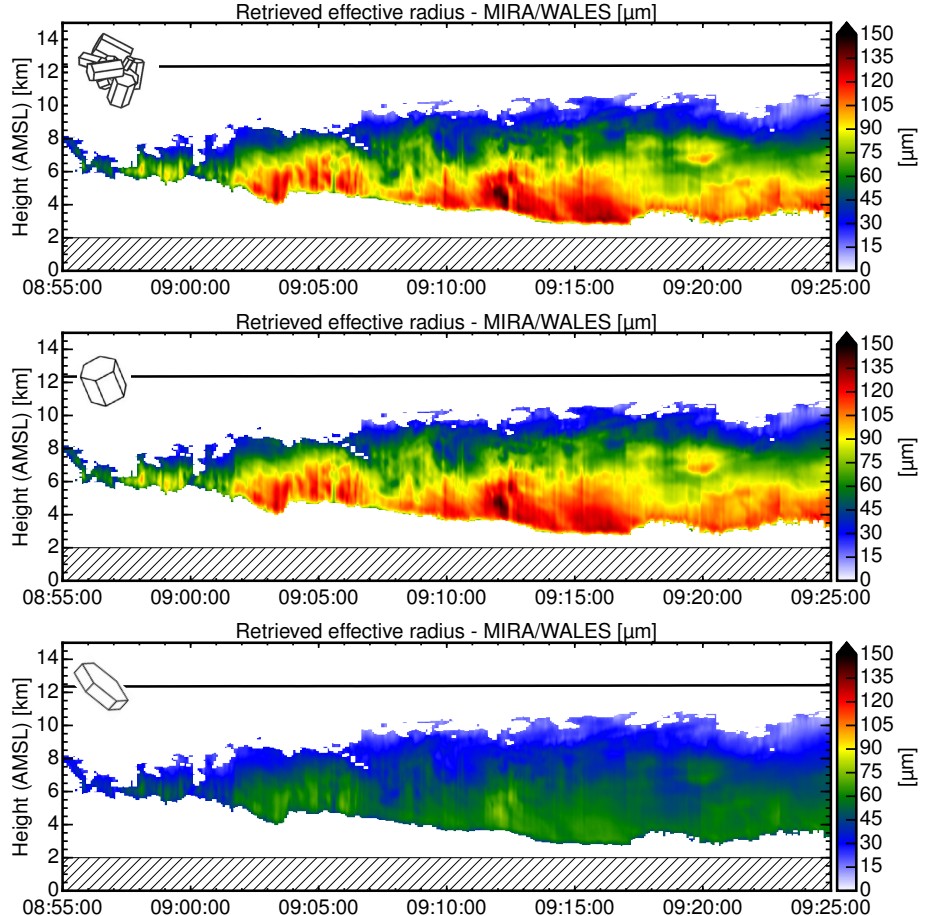

**Figure A1.** Effective radius of ice crystals retrieved by combining information from lidar (Fig. 7b) and radar (Fig. 7c) using the VarCloud framework and the assumption of **(top)** aggregates, **(center)** solid columns and **(bottom)** plates.

*Author contributions.* FE, SG and JD conceived the concept of this study. FE, SG, MW performed the airborne measurements and its
calibration. FE developed the presented methods and carried out the analysis. SG, JD and BM contributed to the interpretation of the results.





SF processed and provided the in-situ data used in this study. FE took the lead in writing the manuscript. All authors provided feedback on the manuscript.

*Competing interests.* The authors declare no conflict of interest.

*Acknowledgements.* This work was jointly supported by the German Aerospace Center (DLR) and the German Research Foundation (DFG)
through the HALO Priority Program SPP 1294, *Atmospheric and Earth System Research with the Research Aircraft HALO (High Altitude and Long Range Research Aircraft)*. The author would like to thank the crew and personnel involved in the NAWDEX campaign. The BAe-146 research aircraft is operated by Airtask and Avalon and managed by the Facility for Airborne Atmospheric Measurements (FAAM). DLR-FX and its pilots are thanked for their great support during planning of research flights. The authors thank Luca Bugliaro for his internal review of the manuscript.



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
