# Peer review of "Why we need radar, lidar, and solar radiance observations to constrain ice cloud microphysics"

_Atmospheric Measurement Techniques, 2020_

## Author Comment (AC1)

**Introduction**

We thank referee #1 for his/her careful reading, comments and suggestions which we address in the following. The authors' answers are printed in italics:

Remark: The figure and page numbers in the referee comments are corresponding to the original manuscript. If not stated otherwise, figure and equation numbers in the authors' answers are referring to the revised, marked-up manuscript version (showing the changes made).

"Review of "Why we need radar, lidar, and solar radiance observations to constrain ice cloud microphysics" by F. Ewald et al. This is a very interesting paper focusing on achieving short-wave radiative closure using multi-instrument cloud observations. The ability of such studies to constrain ice cloud particle habits is particularly noteworthy. The results are especially relevant to upcoming active remote sensing satellite missions (e.g. the ESA/JAXA Earth Clouds and Radiation Explorer mission). The paper is, on the whole, well written and structured. There are, however, some areas where the text must be improved."

→ Thank you very much for your time and effort in compiling this review! We performed all grammatical corrections suggested by the reviewers and revised the mentioned areas to improve their readability and clarity. At the end of this text you will find a detailed track change for the revised manuscript.

**Specific comment**

"P-3: L-70 to 75: This paragraph is confusing (e.g. it is not clear to me at all when you are referring to lidar+radar, passive, or lidar+radar+IR radiometer measurements).
I think the points you are trying to make here are:

1. Combined lidar-radar measurements can provided high-resolution (on the scale of 10s of meters) vertical profiles of cloud properties. This capability can not be matched by passive sensor only based cloud retrievals.

However,

2. Lidar measurements are most sensitive to the particle extinction while radar reflectivity is mostly dependent on the squared-mass distribution.

3. Using lidar derived extinction together with radar reflectivity is not enough to constrain e.g. the effective size or IWC unambiguously. The mapping between the lidar and radar measurements and e.g. IWC and Reff depends significantly on the assumed particle habit and size distribution.

4. Using IR emissivity measurements can help constrain the problem. However, even then ambiguity can remain as IR measurements can saturate with optical depth quite quickly.

Please rewrite the first half of this paragraph (with appropriate references)."

→ Thank you very much for pointing this out. We agree that the original paragraph was confusing since several statements were mixed together. Since we really liked your line of thought, we adopted your suggestion and supplemented it with references.

The first half of the paragraph of section "1.3 Problem Statement" now reads:

Combined radar-lidar measurements can provided high-resolution vertical profiles of cloud properties on the scale of a few dozen meters. This capability can not be matched by cloud retrievals which are based on passive sensor only (Duncan and Eriksson, 2018). However, even the radar-lidar measurements are not enough to constrain ice cloud microphysics, e.g. retrieve the effective radius  $(r_{eff})$  and ice water content (IWC) unambiguously as shown by Ham et al. (2017). While lidar measurements are most sensitive to the particle extinction, radar reflectivity is mostly dependent on the squared-mass distribution of ice particles (Tinel et al., 2005). The mapping between the lidar and radar measurements depends significantly on the assumed particle habit and size distribution (Sourdeval et al., 2018). These assumptions determine the relationship between the extinction and further retrieved quantities like reff and IWC (Cazenave et al., 2019). Here, IR emissivity measurements can help constrain the problem (e.g. Delanoë and Hogan, 2008). But even then, ambiguity can remain as IR measurements saturate quite quickly with optical depth (Hong et al., 2016; Khatri et al., 2018).

**Minor comments**

- "P-1, L-19: Bad sentence: I suggest: "In this case, collocated in-situ measurements indicate that the lack of closure may be linked to unexpectedly high values of the ice crystal number density.""
  - $\rightarrow\,$  Thanks for this suggestion. We adopted your wording which is much more precise.

- "P-2: L-30: "...distribution contribute to...""
  - $\rightarrow$  Correct, the present progressive was not correct here.
- "P-4: L 101: Concluding "..the paper concludes with the presentation of a case...""
  - $\rightarrow$  Thanks, we changed the wording accordingly.
    - P-4, L 101 now reads:

The paper concludes with the presentation of a case with unsuccessful radiative closure which is analyzed and discussed in Sec. 4 using collocated in-situ measurements.

- "P-7: L-177: "...online.." ? Do you mean to say that the lidar forward model you use is being run through a remote web interface ? I think you want to say that the model is available for download. If the later is true, then please give a url, or just delete "online"."
  - $\rightarrow$  You are correct. The referenced model is a standard radiative transfer code. For this reason we deleted the term "online".
- "P-7: L-185: "...and the beam...""

 $\rightarrow$  Thanks, corrected.

- "P-8, First paragraph. Bad first two sentences. I suggest something like

"The ice microphysical and scattering models employed in this study are of central importance. Both the lidar+radar (+radiometer) results as well as the simulated SW radiances used in the closure assessment depend on the ice microphysical and scattering models assumed. In this section, we describe the microphysical and scattering

models employed in this study. We cover both the models/assumptions used in the retrievals and in the simulation of of the short-wave radiances used to assess the closure.""

 $\rightarrow\,$  Thanks for this suggestion! We really like your wording and adopted your suggestion with minor changes.

*P-8, First paragraph now reads:*

The ice microphysical and scattering models employed in this study are of central importance. Both the lidar-radar results as well as the simulated solar radiances used in the closure assessment depend on the ice microphysical and scattering models assumed. In this section, we describe the microphysical and scattering models employed in this study. We cover both, the assumptions used in the retrieval and in the simulation of the solar radiances for the radiative closure.

- "P-8, L 193: "A commonly used framework which simplifies the... is the concept of an effective ice particle density...""
  - $\rightarrow\,$  We changed our wording accordingly.

P-8, L193 now reads:

A commonly used framework which simplifies the variability of naturally occurring ice cloud particles is the concept of an effective ice particle density  $\rho_{i,eff}$ .

- "P-9: L-200: Delete the coma after "Analogous""

 $\rightarrow$  Done.

- "P-10, L-243: Very unclear sentence (e.g. what do you mean by "At first..?". I recommend deleting this sentence."
  - → You are right. We wanted to mention that multilayer cloud scenes pose a problem for the radiative closure. We deleted the first setence and added a short sentence.

P-10, L-243 now reads:

While VarCloud only retrieves properties of ice clouds, solar radiation can also be reflected by liquid water clouds and aerosols. As a consequence, the radiance measurements can contain a mixture of information from ice clouds, underlying water clouds, aerosols, and the surface. This poses a problem for the radiative closure.

- "P-13: L-291: "..scientific objective.." is not the proper phrase here, I recommend simply "target"."
  - $\rightarrow$  Thanks for the suggestion. We changed it to "scientific target".
- "P-13, L-295: Define WCB ?"
  - → WCB stands for "warm conveyor belts". We already defined the term in L-129. Nevertheless, we replaced the abbreviation here as it confused you when reading this paragraph.
- "P-14: L-328 "Overall, the lidar signal is extinguished much more rapidly ... ""

ightarrow Oh, thanks for spotting that.

- "P-17: L-348: "..for all the microphysical models considered...""
  - $\rightarrow$  Thanks, we adopted this wording.
- "P-18: L-365: "Comparison of in-situ and remote-sensing observations""
  - → The titel of section 4 now reads: "Comparison of in-situ and remotesensing observation".
- "P-18: Last line "..and its processing are...""
  - $\rightarrow$  Thanks, changed.
- "P-19: L-393: Define "ICNC""
  - → ICNC stands for "ice crystal number concentrations". We already defined the term in L-181. Nevertheless, we replaced the abbreviation once again at the beginning of this paragraph to recall the term for the reader.
- "P-19: End of page: A small bit of foreshadowing here would help guide the reader, e.g. end the section by stating "The implications of the occurrence of the regions of unexpectedly high ICNCs are discussed in the next section".""
  - $\rightarrow$  Thanks for poiting this out. We adopted your line to guide the reader into the next section.
- "P-20: Last sentence of page: Awkawrd sencence. I suggest "The resulting oversmoothing accross this discontunity could lead to the undesired perturbation of .....""

→ This sentence was indeed a little bit awkward to read. Thanks for the suggestion!

Last sentence on P-20 now reads:

The resulting oversmoothing accross this discontunity could lead to the undesired perturbation of microphysical variables, like the lidar ratio or extinction, in adjacent ice cloud layers.

- "P-21: L-445: "...water vapor absorption which insures that mainly light....""

 $\rightarrow$  We changed this sentence accordingly.

**References**

- Cazenave, Q., Ceccaldi, M., Delanoë, J., Pelon, J., Groß, S., and Heymsfield, A.: Evolution of DARDAR-CLOUD ice cloud retrievals: new parameters and impacts on the retrieved microphysical properties, Atmospheric Measurement Techniques, 12, 2819–2835, doi:https://doi.org/10.5194/amt-12-2819-2019, https://www.atmos-meas-tech.net/12/2819/2019/, 2019.
- Delanoë, J. and Hogan, R. J.: A variational scheme for retrieving ice cloud properties from combined radar, lidar, and infrared radiometer, Journal of Geophysical Research, 113, D07 204, http://www. agu.org/pubs/crossref/2008/2007JD009000.shtml, 2008.
- Duncan, D. I. and Eriksson, P.: An update on global atmospheric ice estimates from satellite observations and reanalyses, Atmospheric Chemistry and Physics, 18, 11205–11219, doi:10.5194/acp-18-11205-2018, https://acp.copernicus.org/articles/18/11205/2018/, publisher: Copernicus GmbH, 2018.
- Ham, S.-H., Kato, S., and Rose, F. G.: Examining impacts of mass-diameter (m-D) and areadiameter (A-D) relationships of ice particles on retrievals of effective radius and ice water content from radar and lidar measurements, Journal of Geophysical Research: Atmospheres, 122, 2016JD025672, doi:10.1002/2016JD025672, http://onlinelibrary.wiley.com/doi/10. 1002/2016JD025672/abstract, 2017.

- Hong, Y., Liu, G., and Li, J.-L. F.: Assessing the Radiative Effects of Global Ice Clouds Based on CloudSat and CALIPSO Measurements, Journal of Climate, 29, 7651–7674, doi:10.1175/JCLI-D-15-0799.1, http://journals.ametsoc.org/doi/abs/10.1175/JCLI-D-15-0799.1, 2016.
- Khatri, P., Iwabuchi, H., and Saito, M.: Vertical Profiles of Ice Cloud Microphysical Properties and Their Impacts on Cloud Retrieval Using Thermal Infrared Measurements, Journal of Geophysical Research: Atmospheres, 123, 5301–5319, doi:https://doi.org/10.1029/2017JD028165, https://agupubs.onlinelibrary.wiley.com/doi/abs/10.1029/2017JD028165, \_\_\_eprint: https://agupubs.onlinelibrary.wiley.com/doi/pdf/10.1029/2017JD028165, 2018.
- Sourdeval, O., Gryspeerdt, E., Krämer, M., Goren, T., Delanoë, J., Afchine, A., Hemmer, F., and Quaas, J.: Ice crystal number concentration estimates from lidar-radar satellite remote sensing Part 1: Method and evaluation, Atmospheric Chemistry and Physics, 18, 14327–14350, doi:https://doi.org/10.5194/acp-18-14327-2018, https://www.atmos-chem-phys. net/18/14327/2018/, 2018.
- Tinel, C., Testud, J., Pelon, J., Hogan, R., Protat, A., Delanoe, J., and Bouniol, D.: The retrieval of ice-cloud properties from cloud radar and lidar synergy, Journal of the Applied Meteorology, 44, 860–875, 2005.

---

## Author Comment (AC2)

**Introduction**

We thank referee #2 for his/her careful reading, comments and suggestions which we address in the following. The authors' answers are printed in italics:

> *Remark: The figure and page numbers in the referee comments are corresponding to the original manuscript. If not stated otherwise, figure and equation numbers in the authors' answers are referring to the revised, marked-up manuscript version (showing the changes made).*

"The Authors discuss the use of passive measurements in the shortwave infrared to provide radiative closure to synergistic retrievals of cirrus cloud microphysical properties utilizing co-located airborne lidar and radar measurements. They discuss two test cases, one where passive measurements show evidence of a good radiative closure and one where they do not. For the latter case, they use co-located in situ observations to relate the lack of radiative closure to an exceptionally high ice crystal number concentration at an altitude of about 7.5 km, which poses challenges to both lidar and radar retrievals.

Overall, I found this paper very interesting and clearly written, and therefore I recommend publication with only minor technical corrections. Specific comments are below:"

> → *Thank you very much for your time and effort in compiling this review! We performed all grammatical corrections suggested by the reviewers and revised the mentioned areas to improve their readability and clarity. At the end of this text you will find a detailed track change for the revised manuscript.*

**Minor technical corrections**

- "Abstract, L19-20. The last sentence was not immediately clear to me when I first read the paper. You may want to replace "narrow" with "attribute" or "relate"."

→ *Thanks for this suggestion. As this sentence was also flagged by referee #1, we adopted their wording: "In this case, collocated in-situ measurements indicate that the lack of closure may be linked to unexpectedly high values of the ice crystal number density."*

– "P2, L56. Do you actually mean "overlapping radar and lidar measurements"?"

→ *Yes, we actually mean the region where radar and lidar data overlap, e.g. the lidar is not yet attenuated while the radar is already sensitive enough. Since the term "overlap" could be confused with the term "lidar overlap", we explain this term now explicitly.*

*P2, L56 now reads:*

*These methods were, however, only applicable to the overlap region where the lidar signal is not yet attenuated but cloud particles are already large enough to be detected by a cloud radar.*

– "P5, Section 2.1. At what altitudes did the aircrafts fly?"

→ *We describe the altitudes and flight patterns in the second paragraph in section "3.2 Case 2: Occluded front clouds":*

*While HALO and the SAFIRE Falcon flew over the cloud layer at an altitude of $13.5\,\mathrm{km}$ and $11\,\mathrm{km}$ respectively, the FAAM BAe-146 performed a profiling flight pattern within the radar-lidar curtain.*

*The BAe-146 profile pattern is shown in Fig. 8d.*

– "P5, L127, "aircraft" -> "aircrafts""

> → *Are you sure about this plural form? We consulted multiple dictionaries which state that "aircraft" does not have a plural form.*

**–** "P6, L151, "spectroradiometer" -> "spectroradiometers""

> → *Ok. Changed this to your suggestion.*

**–** "P7, P179, consider replacing "convergence with the actual measurements" with "convergence of the simulated measurements to the actual ones""

> → *Thanks for this suggestion. Your wording is more natural.*

**–** "P10, L243, "missing" -> "lack of""

> → *You are correct, "lack of" is more appropriate here.*

**–** "P10, L244, "can be reflected by all atmospheric consituents" -> "can also be reflected by liquid water clouds and aerosols"."

> → *Thanks, we adopted your more descriptive wording.*

**–** "P18, last line. Do you mean Fig. 9d instead of 9c?"

> → *Thats correct, thanks for spotting this typo!*